# Upregulation of Protein Synthesis and Proteasome Degradation Confers Sensitivity to Proteasome Inhibitor Bortezomib in Myc-Atypical Teratoid/Rhabdoid Tumors

**DOI:** 10.3390/cancers12030752

**Published:** 2020-03-22

**Authors:** Huy Minh Tran, Kuo-Sheng Wu, Shian-Ying Sung, Chun Austin Changou, Tsung-Han Hsieh, Yun-Ru Liu, Yen-Lin Liu, Min-Lan Tsai, Hsin-Lun Lee, Kevin Li-Chun Hsieh, Wen-Chang Huang, Muh-Lii Liang, Hsin-Hung Chen, Yi-Yen Lee, Shih-Chieh Lin, Donald Ming-Tak Ho, Feng-Chi Chang, Meng-En Chao, Wan Chen, Shing-Shung Chu, Alice L. Yu, Yun Yen, Che-Chang Chang, Tai-Tong Wong

**Affiliations:** 1International Master/Ph.D. Program in Medicine, College of Medicine, Taipei Medical University, Taipei 110, Taiwan; t.minhhuy@gmail.com; 2Department of Neurosurgery, Faculty of Medicine, University of Medicine and Pharmacy at Ho Chi Minh City 700000, Vietnam; 3Graduate Institute of Clinical Medicine, College of Medicine, Taipei Medical University, Taipei 110, Taiwan; abel1063@gmail.com (K.-S.W.); chaomengen@gmail.com (M.-E.C.); a128098527@gmail.com (W.C.); dog52037@gmail.com (S.-S.C.); 4The Ph.D. Program for Translational Medicine, College of Medical Science and Technology, Taipei Medical University, Taipei 110, Taiwan; ssung@tmu.edu.tw (S.-Y.S.); austinc99@tmu.edu.tw (C.A.C.); 5The Ph.D. Program for Cancer Molecular Biology and Drug Discovery, College of Medical Science and Technology and Academia Sinica, Taipei Medical University, Taipei 110, Taiwan; yyen@tmu.edu.tw; 6Joint Biobank, Office of Human Research, Taipei Medical University, Taipei 110, Taiwan; thhsieh@tmu.edu.tw (T.-H.H.); d90444002@tmu.edu.tw (Y.-R.L.); 7Department of Pediatrics, School of Medicine, College of Medicine, Taipei Medical University, Taipei 110, Taiwan; yll.always@gmail.com (Y.-L.L.); minlan456@hotmail.com (M.-L.T.); 8Department of Pediatrics, Taipei Medical University Hospital, Taipei 110, Taiwan; 9Pediatric Brain Tumor Program, Taipei Cancer Center, Taipei Medical University, Taipei 110, Taiwan; 10Department of Radiation Oncology, Taipei Medical University Hospital, Taipei Medical University, Taipei 110, Taiwan; b001089024@tmu.edu.tw; 11Taipei Cancer Center, Taipei Medical University, Taipei 110, Taiwan; 12Department of Medical Imaging, Taipei Medical University Hospital, Taipei 110, Taiwan; kevinh9396@gmail.com; 13Department of Pathology, Wan Fang Hospital, Taipei Medical University, Taipei 110, Taiwan; bluemageh@gmail.com; 14Division of Pediatric Neurosurgery, Neurological Institute, Taipei Veterans General Hospital, Taipei 112, Taiwan; liang4617@hotmail.com (M.-L.L.); roberthhchen3@gmail.com (H.-H.C.); yylee62@gmail.com (Y.-Y.L.); 15Department of Pathology and Laboratory Medicine, Taipei Veterans General Hospital, Taipei 112, Taiwan; diegolin@vghtpe.gov.tw (S.-C.L.); mtho11728@gmail.com (D.M.-T.H.); 16Department of Pathology and Laboratory Medicine, Cheng Hsin General Hospital, Taipei 112, Taiwan; 17Department of Radiology, Taipei Veterans General Hospital, Taipei 112, Taiwan; fcchang@vghtpe.gov.tw; 18Institute of Stem Cell and Translational Cancer Research, Chang Gung Memorial Hospital at Linkou and Chang Gung University, Taoyuan 333, Taiwan; ayu@gate.sinica.edu.tw; 19Genomics Research Center, Academia Sinica, Taipei 115, Taiwan; 20TMU Research Center of Cancer Translational Medicine, Taipei Medical University Taipei, Taipei 110, Taiwan; 21Division of Pediatric Neurosurgery, Department of Neurosurgery, Taipei Medical University Hospital and Taipei Neuroscience Institute, Taipei Medical University, Taipei 110, Taiwan; 22Neuroscience Research Center, Taipei Medical University Hospital, Taipei 110, Taiwan

**Keywords:** Myc-ATRTs, protein synthesis, proteasome degradation, bortezomib, p53

## Abstract

Atypical teratoid rhabdoid tumors (ATRTs) are among the most malignant brain tumors in early childhood and remain incurable. Myc-ATRT is driven by the *Myc* oncogene, which directly controls the intracellular protein synthesis rate. Proteasome inhibitor bortezomib (BTZ) was approved by the Food and Drug Administration as a primary treatment for multiple myeloma. This study aimed to determine whether the upregulation of protein synthesis and proteasome degradation in Myc-ATRTs increases tumor cell sensitivity to BTZ. We performed differential gene expression and gene set enrichment analysis on matched primary and recurrent patient-derived xenograft (PDX) samples from an infant with ATRT. Concomitant upregulation of the Myc pathway, protein synthesis and proteasome degradation were identified in recurrent ATRTs. Additionally, we found the proteasome-encoding genes were highly expressed in ATRTs compared with in normal brain tissues, correlated with the malignancy of tumor cells and were essential for tumor cell survival. BTZ inhibited proliferation and induced apoptosis through the accumulation of p53 in three human Myc-ATRT cell lines (PDX-derived tumor cell line Re1-P6, BT-12 and CHLA-266). Furthermore, BTZ inhibited tumor growth and prolonged survival in Myc-ATRT orthotopic xenograft mice. Our findings suggest that BTZ may be a promising targeted therapy for Myc-ATRTs.

## 1. Introduction

Atypical teratoid/rhabdoid tumors (ATRTs) account for 1–2% of all central nervous system (CNS) tumors in children aged 0–14 years, yet are among the most common malignant CNS tumors in infants less than 1 year old [1]. ATRTs are defined by the loss of INI1 or, rarely, BRG1, encoded by the *SMARCB1* and *SMARCA4* genes, respectively. Patients with ATRTs have dismal outcomes due to their highly malignant nature and young age at diagnosis. There remains no standard therapy for ATRTs [2]. Multimodal treatment strategies include a selective combination of conventional chemotherapy, high dose chemotherapy and stem cell rescue, intrathecal chemotherapy and radiotherapy after tumor resection [2]. The survival rate, even with aggressive treatment, is still low (2-year survival rate is 32.6–44.6%) [3]. Moreover, currently used cytotoxic therapies incur some neurocognitive side effects, particularly in infants, highlighting the urgent need for novel targeted therapies.

One target for cancer therapy is the ubiquitin–proteasome pathway (UPP), which plays the principal role in intracellular protein degradation [4]. UPP maintains cellular proteostasis and regulates multiple intracellular processes, including cell cycles, DNA repair and apoptosis [5]. Therefore, proteasome inhibitors cause an accumulation of protein substrates and dysregulation of cellular proteostasis, leading to apoptosis in cancer cells [6]. Bortezomib (BTZ) (PS-341), a first-generation proteasome inhibitor, is a well-established targeted therapy in multiple myeloma (MM) [7,8] and mantle cell lymphoma [9]. In MM, the protein synthesis rate is correlated to its sensitivity to BTZ [10,11].

ATRTs are classified into three epigenetic subgroups, including ATRT-SHH, ATRT-TYR and ATRT-MYC [12,13]. Myc-ATRTs (identified by the overexpression of *Myc* oncogenes) have the worst prognosis [12,13]. *Myc* is a key factor in controlling translation and inducing protein synthesis in cancer cells [14,15]. In this study, we established a matched PDX model from an infant who was diagnosed with ATRT with two recurrences. RNA sequencing (RNA-seq) analysis revealed that the molecular profiles of the primary and recurrent tumors shift from the SHH to the Myc subgroup. Additionally, protein synthesis and the expression of proteasome components were increased in the recurrent tumors. We hypothesized that protein synthesis and proteasome degradation might be upregulated and associated with malignancy, providing a therapeutic target for Myc-ATRTs.

## 2. Results

### 2.1. Establishing a Matched Model for the Primary and Recurrent Atypical Teratoid Rhabdoid Tumors

To establish the ATRT model, we utilized samples obtained from an infant (TM71) who was diagnosed with supratentorial ATRT at age eight-months. This patient had undergone three operations for tumor resection. Whole-exome sequencing (WES) from blood and the primary tumor revealed a somatic nonsense mutation in *SMARCB1* (exon2: c.157C > T, p.53R > X). We generated six passages of the primary PDX mice, six passages of the first recurrent PDX mice and three passages of the second recurrent PDX (Figure 1a). We also created a continuous cell line, Re1-P6, from the sixth passage of the first recurrent PDX tumor (Appendix A). To test the tumorigenic potential of Re1-P6 cells, we orthotopically implanted Re1-P6 cells (4 × 10^5^ cells/10 µL) into the cerebrum of 6–8 week-old NOD.CB17-Prkdc^scid^/NcrCrl (NOD/SCID) mice. The Re1-P6 cells retained malignancy with a tumor formation rate of 100% (8/8) after 21-days-post transplantation (dpt) (Appendix A). Loss of INI1 in the tumors of the mice was confirmed by immunohistochemistry (IHC) (Figure 1b). 

We then used RNA sequencing (RNA-seq) data to identify molecular subgroups of primary and recurrent tumors based on 39 subgroup genes [12]. We discovered an association between primary tumors and the SHH subgroup, in contrast to recurrent tumors, which transformed into the Myc subgroup (Figure 1c). Additionally, the molecular profiles of the P1–P6 samples of the first and the P1–P3 samples of the second recurrent PDX models were found to be highly similar to the P0 samples of the second recurrent patient sample (Figure 1c), suggesting that the malignant transformation in vivo might recapitulate the tumorigenic process in the clinical samples.

To assess the similarity of tumor samples, we used RNA-seq data for clustering. The principal components analysis plot showed that the primary and recurrent PDX samples were separated into two subgroups (Figure 1d). We further used DeSeq2 [16] to analyze the differential gene expression between two groups: group 1, which includes the first three passages of the primary PDX samples (*n* = 3) and group 2, which includes the first three passages of the first and the second recurrent PDX samples (*n* = 6). Next, we put the ranked gene list into the gene set enrichment analysis (GSEA) [17]. We identified the *Myc* gene set was upregulated in recurrent tumors with NSE = 1.9, FDR = 0.22, whereas the SHH gene set was prominent in the primary tumors with NSE = −1.85, FDR = 0.2 (Figure 1e,f). These data confirmed that the molecular subgroups of recurrent tumors changed from SHH to Myc.

### 2.2. Upregulation of Protein Synthesis and Proteasome Degradation in Recurrent ATRT

We found that recurrent tumors showed a higher proliferation rate than the primary tumors, assessed via Ki67 stains (Appendix A). Recurrent PDX mice also had a higher tumor growth rate than the primary PDX mice (33.43 ± 3.51 vs. 10.64 ± 1.75 mm^3^/day, *p* < 0.0001, *t*-test), suggesting recurrent tumors harbored more malignant characteristics than the primaries. To identify genetic factors that drive malignancy, we looked for signaling pathways that were upregulated in recurrent tumors. The GSEA showed robust upregulation of gene sets involved in protein synthesis and ribosome biogenesis in recurrent tumors, including Reactome translation (NES = 2.17, FDR = 0.001), Reactome eukaryotic translation initiation (NES = 2.1, FDR = 0.001), Reactome SRP-dependent co-translational protein targeting to membrane (NES = 2.2, FDR = 0.001), Reactome eukaryotic translation elongation (NES = 2.17, FDR = 0.002), Reactome eukaryotic translation termination (NES = 2.14, FDR = 0.002), GOBP Mitochondrial translation (NES = 1.95, FDR = 0.15), Reactome rRNA processing (NES = 2.14, FDR = 0.002), Reactome rRNA processing in the nucleus and cytosol (NES = 2.19, FDR = 0.001) and GOBP rRNA metabolic process (NES = 1.96, FDR = 0.015) (Figure 2a,b). We then confirmed the increase in protein synthesis in the patient’s recurrent tumors. The RNA-seq heatmap showed greater expression of the Reactome translation gene set in the patient’s recurrent tumors than in the patient’s primary tumors (Figure 2c). 

Upregulation of protein synthesis may place a greater burden on the protein degradation system. Therefore, we tested whether proteasome degradation is also activated in recurrent tumors. We found increased expression of the proteasome-encoding gene in recurrent tumors compared with the primary tumors in ATRT PDX models and patient samples (Figure 2d, Appendix A). We further compared proteasome-encoding gene expression in 24 human ATRT samples (22 primary ATRTs and two recurrent ATRTs) (Appendix A) with four normal brain samples. Raw RNA-seq data of four normal brain tissues were downloaded from the GEO data set (GSM2501173, GSM2501174, GSM2501175 and GSM2193194) (Appendix A). We found that the expression of 34 of the 42 proteasome-encoding genes was significantly higher in tumor samples than in normal brain tissues (Figure 2e). Furthermore, we investigated the expression of proteasome-encoding genes in 4 ATRT public data sets [13,18,19,20] compared with that of normal CNS tissues [21] by using the R2 Platform (http://r2.amc.nl). Similarly, tumor samples expressed higher levels of proteasome coding genes (38 of 42 genes) than did the normal CNS tissues (Appendix A). 

To confirm the association between increased expression of proteasome-encoding genes and malignant behavior, we stained patient samples for PSMD4 and PSMB4, which represent 19S and 20S subunit encoding genes, respectively. We observed higher expression of PSMD4 and PSMB4 protein levels in the recurrent tumors compared with the primary tumor (Figure 2f,g) and normal brain tissues (Appendix A). Overall, these data suggest that proteasome-encoding genes are more prominent ATRTs, particularly in recurrent tumors, than in normal CNS tissues and may be positively correlated to the malignancy of ATRTs.

### 2.3. Importance of PSMD4 and PSMB4 in ATRT Cell Survival

To determine whether PSMD4 and PSMB4 are essential for ATRT cell survival, we knocked down PSMD4 and PSMB4 expression in Re1-P6 and BT-12 cells using shRNA. With the MTT assay, we found that decreased expression of PSMD4 or PSMB4 results in decreased proliferation of Re1-P6 and BT-12 cells (Figure 3b,e,h,k). Additionally, using colony formation assays, we found that reducing PSMD4 or PSMB4 expression has a long-term effect on the proliferation of Re1-P6 cells and BT-12 cells (Figure 3c,f,i,l). We further analyzed the dependency scores of all proteasome encoding genes in five ATRT cell lines (BT-12, BT-16, CHLA06ATRT, CHLA266 and COGAR359) from the project DepMap-Achilles Public 19Q2 dataset [22]. We found that the majority of proteasome coding genes (37/42 genes) contribute to the survival of tumor cells with dependency scores of less than 0 (Appendix A). This suggests that proteasome encoding genes are essential for ATRT cell survival. 

### 2.4. Proteasome Inhibitor BTZ Inhibited Tumor Growth and Induced Apoptosis In Vitro

We hypothesized that the upregulation of protein synthesis, which is a downstream target of *Myc* [14,15], might make the tumor cell more sensitive to proteasome inhibitors. Therefore, we included Re1-P6 cells and two other Myc-ATRT human cell lines (BT-12, CHLA266) [12] in the in vitro drug test. We compared the antitumor effect of BTZ with that of other targeted drugs, including the aurora kinase A inhibitor alisertib (MLN8237) (in a phase 2 clinical trial for patients with ATRT) (NCT02114229), histone deacetylase inhibitor SAHA (enhances the effect of ionizing radiation on ATRT cells) [23] and multiple kinase inhibitor lenvatinib because of the high expression of *FGFR1* and *RET* mRNA in our PDX models (Appendix A).

BTZ demonstrated a potential effect on three cell lines with a half-maximal inhibitory concentration (IC_50_) ranging from 5.84 to 8.7 nM (Figure 4a) and IC_90_ ranging from 10.6 to 16.4 nM, concentrations that are achievable in clinical practice; IC_90_ values of alisertib, SAHA and lenvatinib were 41.3–558 µM, 24.2–822.91 µM and 67.7–1242.1 µM, respectively (Figure 4b–d, Appendix A). Studies have shown that BTZ inhibits tumor growth and induces apoptosis in human MM cells [24]. To evaluate the effect of BTZ on apoptosis, we stained the Re1-P6, BT-12 and CHLA266 cells with propidium iodide following 24- or 48-hour exposure to BTZ (10–20 nM) and used flow cytometry to assess the cell cycle. We found that BTZ dose and time-dependently increased the sub-G1 population, which represents apoptosis cells, in three ATRT cell lines (Figure 4e, Appendix A). Furthermore, immunoblotting for cleaved caspase-3 confirmed the time-dependent increase of the apoptosis marker in BTZ-treated ATRT cells (Figure 4f, Appendix A). The mechanism of BTZ is suggested to be through the inhibition of proteasome function, which leads to the accumulation of protein substrates such as p53 and IκBa [24,25]. We first checked the proteasome activity of ATRT cells after 4–20 h of BTZ exposure. We found that BTZ inhibited proteasome activity in treated cells (Figure 4g, Appendix A). Furthermore, immunoblotting showed that BTZ caused the accumulation of p53 as well as polyubiquitinated p53 in treated cells (Figure 4f, Appendix A). We also investigated the NFκB signaling pathway in BTZ- treated ATRT cells. However, no accumulation of IκBa in ATRT cells after treatment with BTZ was observed (Appendix A), suggesting that BTZ induces apoptosis in Myc-ATRT through the p53 pathway but not through the NFκB signaling pathway.

Furthermore, we compared the cytotoxic effect of BTZ on two SHH-ATRT cell lines (CHLA-02 and CHLA-04) and murine fibroblast L929 cell line as controls. The IC_50_ of BTZ on CHLA-02, CHLA-04 and L929 were 15.1 (14.3–15.9) nM, 15.8 (14.5–17.3) nM and 38.3 (34.9–42) nM, respectively (Figure 4a, Appendix A). The less sensitivity of SHH-ATRT cell lines to BTZ compared with Myc-ATRT further supported the correlation between Myc activity and the sensitivity to BTZ in ATRTs. 

### 2.5. BTZ Inhibited Tumor Growth and Prolonged Survival Time in the Orthotopic Xenograft Model

We explored the antitumor effect of BTZ in the orthotopic human Myc-ATRT xenograft model injected with Re1-P6 cells (Figure 5a). Although BTZ treatment did not notably affect body weight (Figure 5b), it significantly prolonged median survival in the treated group (*N* = 13) compared with the control group (*N* = 12) (58 vs. 41 days, *p* < 0.0001) (Figure 5e). BTZ also clearly inhibited tumor growth in the treated group (*P* = 0.0024, Mann–Whitney U test) (Figure 5c,d, Appendix A). We next investigated whether BTZ induced the accumulation of p53 in treated mice; p53 IHC staining in mice brain tumors showed that the ATRTs in the treated group had a higher expression of p53 than the tissues in the control group (Figure 5f). This finding suggests that the antitumor effect of BTZ is associated with the accumulation of p53. 

Studies have identified p53 status as a key determining factor of the sensitivity of tumor cells to BTZ.^32-34^ We investigated p53 status in ATRTs. IHC results revealed a high expression of p53 in primary and recurrent tumors (Figure 6a). The WES of 16 patients with ATRT identified all cases as *TP53* wild-type (Figure 6b). Additionally, ATRTs had higher expression of *TP53* mRNA than normal brain tissues (Figure 6c). Our data identified ATRTs as p53-proficient tumors and this characteristic likely contributed to the increased sensitivity to BTZ.

## 3. Discussion

We demonstrated that protein synthesis and proteasome degradation were upregulated in the Myc-ATRT model, which led to sensitivity to the proteasome inhibitor BTZ. The in vitro drug test showed that Myc-ATRT cell lines were more sensitive to BTZ (IC_50_ = 5.84– 8.7 nM) than SHH-ATRT cell lines (IC_50_ = 15.1–15.8 nM). The sensitivity of Myc-ATRT cells to BTZ was also comparable with multiple myeloma cells [24], in which disease that BTZ has been considered as a first-line treatment [8]. We also demonstrated that BTZ inhibited tumor growth and induced apoptosis through the accumulation of p53. 

The *Myc* oncogene promotes protein synthesis by directly regulating translation [14,15]. Our findings revealed that the Myc signaling pathway and protein synthesis were simultaneously upregulated in recurrent ATRTs in both the patient samples and the PDX model. These results further support the link between *Myc* and protein synthesis in ATRT cells. Moreover, the correlation of Myc activity and *SMARCB1*-deficient cancers has been identified in priors studies [26,27,28]. Alimova et al. found that *SMARCB1*-deficient ATRTs expressed higher Myc activity compare to normal brain tissues, regardless of the subtypes of ATRTs [28]. Carugo et al. reported that the Myc-p53 axis regulates cellular proteostasis in *SMARCB1*-deficient malignant rhabdoid tumors [27]. Additionally, the SNF5 subunit encoded by *SMARCB1* has been identified as an inhibitor that prevents the DNA-binding ability of *Myc* [26]. Together, these findings suggest that *SMARCB1* deficiency is associated with an increased rate of protein synthesis and Myc activity in ATRTs, particularly in Myc-ATRTs, which may explain their high sensitivity to BTZ. 

Proteasome inhibitor BTZ is known to be effective in the treatment of MM [7,8] and mantle cell lymphoma [9]; however, thus far, these results have not been duplicated in solid tumors [29]. One hypothesis for the sensitivity of malignant plasma cells to BTZ is the loss of balance between the proteasome load and capacity [4]. Malignant plasma cells are characterized by exaggerated protein synthesis, which creates a significant burden for the proteasome degradation system. This feature of malignant plasma cells identifies them as the cancer cells that are the most sensitive to proteasome inhibitors. The rate of protein synthesis is considered the key determinant of the sensitivity of malignant plasma cells to proteasome inhibitors [10,11]. In our study, protein synthesis and proteasome degradation were enhanced in the Myc-ATRT PDX models, which may have made them sensitive to BTZ (Figure 5g).

BTZ inhibits the function of the proteasome system, thus stabilizing the protein substrates. Two main protein substrates of the proteasome that are involved in the antitumor effects of BTZ are IκBa [24,30,31] and p53 [32,33,34]. We found that BTZ induced apoptosis and inhibited Myc-ATRT cell growth through the accumulation of p53 and not by stabilizing IκBa. P53 activation is associated with increased sensitivity of tumor cells to BTZ [35,36,37]. *TP53* RNA was highly expressed in ATRTs compared with normal brain tissues and no patient in our cohort harbored a *TP53* mutation. These results are consistent with those of studies that reported the overexpression of p53 in 70% of ATRTs [38] and a low rate of *TP53* mutations harbored by ATRTs [13]. Additionally, activation of p53 was observed in the context of *SMARCB1* deficiency [27]. These findings imply that activation of p53 in ATRTs may contribute to their increased sensitivity to BTZ.

A recent study by Hong et al. demonstrated that genes related to the ubiquitin–proteasome system are necessary for the survival of renal medullary carcinoma cells, which share a molecular profile with *SMARCB1*-deficient cancers [39]. Increased sensitivity to BTZ in renal medullary carcinoma and other *SMARCB1*-deficient cells, including MRTs (Malignant rhabdoid tumors) and ATRTs, might be dependent on the loss of *SMARCB1* [39]. A similar result was observed in *SMARCB1*-deficient MRT models—loss of *SMARCB1* activates unfolded protein response and increases the endoplasmic reticulum stress, leading to enhanced sensitivity of proteasome inhibitors in *SMARCB1*-deficient MRT [27]. Recently, molecular similarity has been identified between Myc-ATRTs and MRT [40]. Our findings demonstrate the efficacious antitumor effects of BTZ in Myc-ATRT cells and orthopedic xenograft Myc-ATRT and strongly support the sensitivity to BTZ in *SMARCB1*-deficient cancers, particularly the Myc subgroup. 

In this study, most of the proteasome-encoding genes are the essential constituents of the proteasome in ATRTs cells for cell survival, with the exception of *PSMB8*, *PSMB9*, *PSMB10*, *PSMD5*, *PSMD9* and *PSME4* (Appendix A). *PSMB8*, *PSMB9* and *PSMB10* are three immunoproteasome genes that play a critical role in generating the antigenic peptides [41]. The expression of immunoproteasome is significantly upregulated in some cancer cells that endogenously or exogenously induce interferon-gamma (IFN-γ) [42]. However, we found the expression of *PSMB8*, *PSMB9* and *PSMB10* were not significantly different between ATRTs (4 public data sets [13,18,19,20]) and normal central nervous system (CNS) tissue [21] (Appendix A). A similar result was observed in our cohort of 24 ATRTs samples. There is no significant difference in *PSMB10* expression in ATRTs and normal brain tissues (Figure 2e). Proteasome β subunits, β7, β5, β2 and β1 (encoded by the *PSMB4*, *PSMB5*, *PSMB7* and *PSMB6* genes) are the essential constituents of 20S core of the 26S proteasome in ATRTs cells. BTZ selectively binds to the β5 subunit and interacts with the β1 subunit, thereby inhibiting ubiquitinated proteolysis [43,44]. Additionally, proteasomal assembly requires the C-terminal tail of *PSMB4* to intercalate into a groove between the *PSMB6* and *PSMB7* subunits [45]. These interactions with BTZ contributed to interfering with the assembly of the 20S proteasome may allow a therapeutic activity of BTZ in ATRT.

## 4. Materials and Methods 

### 4.1. Cells Cultures 

Human ATRT cell lines, BT-12, CHLA-266, CHLA-02 (ATCC^®^ CRL-3020™) and CHLA-04 (ATCC^®^ CRL-3036™) have been described previously [46,47,48]. BT-12 and CHLA-266 are gifts from the Children’s Oncology Group. CHLA-02 and CHLA-04 were generously provided by Dr. Annie Huang (University of Toronto, Toronto, ON, Canada). BT-12 and CHLA-266 cells are Myc-ATRT (Group 2 ATRT), while CHLA-02 and CHLA-04 cells are SHH-ATRT (Group 1 ATRT) [12]. The Re1-P6 ATRT cell line was primarily cultured from the sixth passage of the first recurrent PDX tumor. BT-12, CHLA-266 and L929 cells (ATCC^®^ CCL-1™) were maintained in a base medium of Iscove’s Modified Dulbecco’s Medium (Thermo Scientific, Grand Island, NY, USA) containing 20% fetal bovine serum (FBS) (Thermo Scientific, Grand Island, NY, USA) CHLA-02 and CHLA-04 cells were cultured in Dulbecco’s Modified Eagle Medium (DMEM): F12 (Thermo Scientific, Grand Island, NY, USA) with 20 ng/mL human recombinant basic fibroblast growth factor and B 27 supplement. Re1-P6 cells were grown in DMEM (Thermo Scientific, Grand Island, NY, USA) with 5% FBS. All the cell lines were incubated at 37 °C in an atmosphere of 5%CO2. 

### 4.2. Cell Growth Inhibition Assay 

Cells were harvested at a confluent rate of 80–90% and seeded into 96-well plate with 5000 (L929), 10.000 (BT-12, CHLA-02, CHLA-04) and 15.000 (CHLA-266, Re1-P6) cells per well. Cell growth inhibition assay was assessed after 72 h of exposure to a serial dilution of BTZ (2.8–100 nM) (Selleckchem, Houston, TX, USA), alisertib (MLN8237) (0.2–200 µM) (Selleckchem), suberoylanilindehydroxamic acid (SAHA) (2.6–100 µM) (Selleckchem), lenvatinib (0.1–100 µM) (Selleckchem, Houston, TX, USA). Proliferation assay was performed by incubating the cells with MTT 5mg/mL (Bionovas, Taipei, Taiwan) for adherent cells or with MTS (ab197010, abcam) for suspension cells at 37 °C for 4 h. Then dark blue formazan crystals were solubilized in dimethyl sulfoxide (DMSO) (Bionovas, Taipei, Taiwan) for MTT assay. The formazan dye’s absorbance was measure at the optical density of 570 nm for MTT and of 490 nm for MTS with a spectrophotometer (SpectraMax 190, Molecular devices, San Jose, CA, USA). All experiments were triplicated in two independent biological replicates.

### 4.3. Colony Formation Assay 

Cells were seeded into 6-well plates with 2000 and 4000 cells per well for BT-12 and Re1-P6, respectively. Colonies formed after 10 days (BT-12) and 20 days (Re1-P6) were fixed with 100% methanol and stained with 0.5% crystal violet (Merck) in 25% methanol. 

### 4.4. Apoptosis Assay 

Tumor cells were harvested after 24- and 48-hour exposure to BTZ at concentrations of 10, 15, 20 nM. After being fixed with 75% ethanol, the cells were stained with propidium iodide (Thermo Scientific) for flow cytometry (Attune™ NxT Flow Cytometer, Thermo Fisher Scientific, Waltham, MA, USA) to assess the cellular DNA contents. 

### 4.5. Immunohistochemistry 

ATRT tissues from patient, PDX model and orthotopic xenograft mice model were embedded in paraffin after fixing in 4% paraformaldehyde. Normal brain tissue slides (NBP2-50617, Novus Biologicals, Centennial, CO, USA) were used as controls. Tumor sections (5 µm in thickness) were deparaffinized twice in xylene for 10 min each and twice in ethanol for 2 min each. Sections were fixed in 4% paraformaldehyde for 20 min, air dried and blocked with 10% goat serum containing 1% BSA in PBS at room temperature for 30 min. The following primary antibodies were used for IHC: anti-INI1 (1:300, 612110, BD Biosciences, San Jose, CA, USA), anti-PSMD4 (1:100, HPA038807), anti-PSMB4 (1:25, HPA006700, Atlas Antibodies, Stockholm, Sweden), anti-p53 (1:600, MS-187-P0, Thermo Scientific, Fremont, CA, USA) and anti-Ki67 (1:800, M7240, Agilent, Santa Clara, CA, USA). Appropriate positive and negative controls were included in the IHC staining. Images were captured and analyzed by TissueFAXS (TissueGnostics, Vienna, Austria) at the Taipei Medical University (TMU) Core Facility Center (Taipei, Taiwan).

### 4.6. Western Blots 

Whole-cell lysates were harvested using HyTra Cell Protein Extraction Reagent (Goal Bio, Taipei, Taiwan) containing Proteinase Inhibitor Cocktail (Bionovas, Taipei, Taiwan) and immunoblotted. The protein lysates were resolved on 10% polyacrylamide sodium dodecyl sulfate (SDS) gels. The following antibodies were used for immunoblotting—anti-IκBα (L35A5), anti–NF-κB p65(D14E12), anti-Phospho-NF-κB p65 (Ser536) (93H1) and anti-cleaved caspase-3 (Asp175) (5A1E) were from Cell Signaling (Danvers, MA, USA); anti-p53 Ab-6(Clone DO-1) was from Thermo scientific (Fremont, CA, USA); anti-PSMD4, anti-PSMB4 and anti-ACTB were from Atlas Antibodies (Stockholm, Sweden). 

### 4.7. Immunoprecipitation

Three ATRT cell lines (Re1-P6, BT-12 and CHLA-266) were harvested after being cultured for 4, 8, 12, 16 and 20 h in the presence of BTZ (20 nM). Cell lysate samples were incubated with 50 µL of protein A/G beads and anti-p53 antibody (Cell signaling, Danvers, MA, USA) at 4 °C for 3 h under gentle rotation. The resultant protein complex was boiled in the sample buffer containing SDS and analyzed using immunoblotting with anti-Ubiquitinylated proteins antibody (04-263, EMD Millipore, Temecula, CA, USA) and anti-p53 antibody (Cell signaling, Danvers, MA, USA). 

### 4.8. Proteasome Activity Assay

ATRT cells (10^6^ cells) were incubated with 20nM BTZ for 4, 8, 12, 16, 20 h. Whole-cell lysate (40 µg protein) was incubated with substrate Suc-Leu-Leu-Val-Tyr- 7-Amino-4-methylcoumarin (LLVT-AMC) (CHEMICON^®^’s 20S proteasome activity assay kit) at 37 °C for 2 h. The released AMC fluorescence was quantified using a 380/460 nm filter set in a fluorometer (SpectraMax M2e, Molecular devices, San Jose, CA, USA). 

### 4.9. Lentiviral Transduction

Short hairpin RNA *PSMD4* and *PSMB4* were purchased from the National RNAi Core Facility (Academia Sinica, Taiwan). pLKO.1-Sh1-*PSMD4* and Sh2-*PSMD4* target the sequences CCGACAAGGCAAGAATCACAA and GCACGGAATATAGGGTTAGAT of *PSMD4*, respectively. pLKO.1-Sh1-*PSMB4* and Sh2-*PSMB4* target the TGCGAGTCAACAACAGTACCA and GAGAGAGCTTCCTCGGTTATG of *PSMB4*, respectively. Re1-P6 and BT-12 cells were infected with lentiviral particle contained ShRNA-*PSMD4* or ShRNA-*PSMB4* construct and 8 µg/mL polybrene (Sigma-Aldrich) for 24 h. After the selection of 2 µg/mL puromycin (Sigma-Aldrich, St. Louis, MO, USA) in 48 h, the cells were maintained in 5% FBS DMEM with 1 µg/mL puromycin to obtain the stable knock-down cell lines. Re1-P6 and BT-12 cells transfected with empty pLKO.1 vectors were used as the control group.

### 4.10. Establishing PDX and Orthotopic Human ATRT Xenograft Models 

The surgical specimens were obtained from an infant with ATRT (TM71), following the protocol of the TMU institutional review board. Consent from the family was obtained for using the sample tissue for PDX and orthopedic xenograft models. This research has been approved by TMU ethic committee for ATRT PDX model on 01 January 2018 (ethic code: LAC2017-0071) and for ATRT orthotopic xenograft mice model on 01 August 2018 (ethic code: LAC-2017-0425). The tumor samples were subcutaneously implanted into flanks of 6–8-week-old NOD/SCID mice to generate the PDX models. For orthotopic human ATRT xenograft models, Re1-P6 cells were suspended in DMEM and Matrigel^TM^( Life Science, Tewksbury, MA, USA) with a ratio of 2:1 (4 × 10^5^ cells/10 µL) and injected into the right cerebral hemisphere of 6–8 week-old NOD/SCID mice (2.5 mm lateral, 0.14 mm posterior to the bregma and 3 mm in depth) [49]. A pre-treatment brain MRI was performed at 21 dpt to ensure tumor formation (7-T/40-cm magnet, a Biospec Bruker console). BTZ was dissolved in DMSO (2%) and diluted in phosphate buffer solution (PBS) and polyethylene glycol (PEG) 300 (30%). The treatment was started at 22 dpt with 0.4 mg/kg BTZ through tail vein injection twice-weekly for 2 weeks. The control group was treated using a PBS with DMSO (2%) and PEG300 (30%). Posttreatment brain MRIs were obtained 3–7 days after the last dose (35–40 dpt). Neurological deficits and body weight were monitored daily and tumor volumes were calculated based on post-contrast T1-weighted sequences using Image J (http://rsb.info.nih.gov/ij).

### 4.11. RNA-Seq Analysis, Gene Set Enrichment Analysis and Whole-Exome Sequencing Analysis

For RNA-Seq analysis, total RNA was purified using the RNeasy Mini Kit (Qiagen, Valencia, CA, USA). Stranded mRNA libraries were generated by using an Illumina TruSeq Stranded mRNA Library Prep Kit (Illumina). 60 ng/µL poly(A)-selected RNA from each sample was run in two lanes of a Nextseq^®^ 500 System (Illumina) for 150 cycles of multiplexed paired-end reads. Pseudoalignment RNA quantification was performed using Kallisto [50] with default settings, aligning against the human genome build GRCh38. Gene expression table were extracted by tximport [51] and normalized by variance stabilizing transformation (VST) method with DESeq2 [16] in R environment. Clustering analysis was performed on the basis of published 39 subgroup-specific signature genes [12] expression levels using the consensus clustering default parameters through principal component analysis in R environment. Differential gene analysis was performed using the DESeq2 with log fold change >0 and adjusted *p* value < 0.05. Gene set enrichment analysis (GSEA) was performed using GSEA Desktop software [17]. Visualization of the significantly enriched pathway (FDR < 0.05) was created using Cytoscape application [52]. For identifying somatic mutations, raw paired-end reads from WES generated by Nextseq^®^ 500 System were aligned to the human genome build hg38 using Burrows–Wheeler aligner [53]. Mutations calling and filtering were performed using Mutect2 and FilterMutectCalls tool with default parameters in GATK and mutations in IGV with alignment level were visualized [54]. Common single-nucleotide polymorphisms (SNP) were removed by referenced dbSNP (build 151) and candidate mutations were filtered using ANNOVAR [55] with the ClinVar and COSMIC databases [56]. Annotated mutations were further filtered by MAF < 0.01 by using ExAC database [57]. 

### 4.12. Sanger Sequencing

Genomic DNA was extracted from Re1-P6 cells using cell lysis solution and Proteinase K (Qiagen) according to manufacturer’s instruction. Validation of *SMARCB1* mutation was performed by bidirectional Sanger sequencing, following standard protocol. The forward (5′-CCCTTCCCTGTGGTGCTG-3′) and reverse (5′-TCATGACATAAGCGAGTGGTT-3′) primers were designed to amplify a 249 bp product and detect C157T mutation in *SMARCB1*, respectively. PCR was performed in 25 mL reaction mixtures containing 12.5 µL of Solg™ h-Taq DNA Polymerase Smart mix (SolGent, Daejeon, Korea), 5 µM concentrations of each primer and 1 µL (20 ng) of sample gDNA. The PCR condition was performed on the GeneAmp^®^ PCR System 9700 (Applied Biosystems, Waltham, MA, USA) and consisted of 15 min at 95 °C for initial denaturation and 40 cycles of 95 °C for 20 s, 60 °C for 40 s and 72 °C for 30 s and 5 min at 72 °C for final extension. Sequencing data were compared to the annotated by wild-type sequence. 

### 4.13. Data Availability 

RNA-seq data are available in Gene Expression Omnibus (GSE140195).

### 4.14. Statistical Analysis 

Statistical analysis was performed with Prism 7.0 software (GraphPad, version 7.0, San Diego, CA, USA) and the tests are reported in figure legends. *p* ≤ 0.05 was considered statistically significant. 

## 5. Conclusions

ATRTs are the most malignant brain tumors in early childhood and remain incurable. In this study, we found that Myc-ATRT models demonstrate upregulation of protein synthesis and proteasome degradation. Tumors express higher levels of proteasome encoding genes that are vital for their survival. BTZ targets proteostasis, inhibiting tumor growth and inducing apoptosis via the accumulation of p53 in three Myc-ATRT cell lines and orthotopic graft mice. Our findings suggest BTZ may be a promising targeted therapy for Myc-ATRTs.

## Figures and Tables

**Figure 1 cancers-12-00752-f001:**
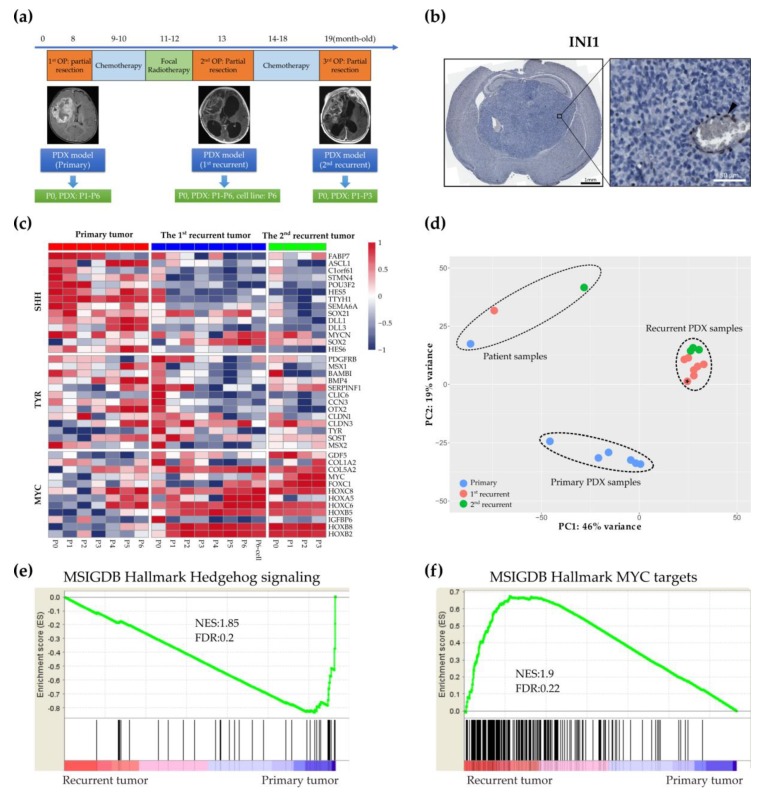
Establishing paired models for Atypical teratoid rhabdoid tumor (ATRT). (**a**) Paired patient-derived xenograft (PDX) models were generated from three surgical samples of one patient with ATRT. The Re1-P6 continuous cell line was created from the sixth passage of the first recurrent PDX tissue. (**b**) Representative immunohistochemistry (IHC) images indicated the loss of IN1 in brain tumors of orthotopically xenograft mice (Re1-P6 cells). Vascular endothelial cells were used as a positive control (black arrowhead). Scale bar, 1 mm (left panel), 50 µm (right panel). (**c**) Gene expression heatmap of the patient, PDX tissues and Re1-P6 cells. The molecular subgroup changed from the ATRT-SHH subgroup in primary tumors to the Myc subgroup in recurrent tumors. P0: Patient samples, P1–P6: PDX samples. (**d**) Principal component analysis categorized PDX tumor samples into three groups, group 1 (patient samples), group 2 (primary PDX samples) and group 3 (recurrent PDX samples, including Re1-P6 cells (*)). (**e**,**f**) GSEA of the primary and recurrent PDX samples revealed upregulation of the SHH signaling pathway in primary tumors, with NSE = −1.85 and FDR = 0.2, (**e**) and of the Myc signaling pathway in recurrent tumors, with NSE = 1.9 and FDR = 0.22 (**f**).

**Figure 2 cancers-12-00752-f002:**
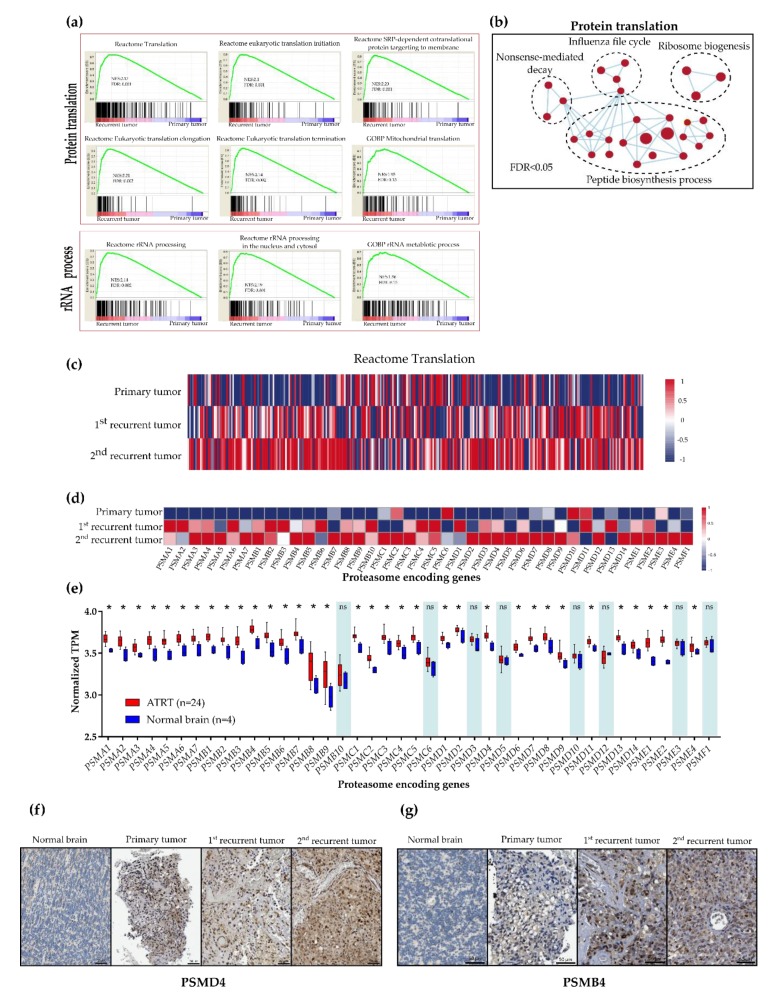
Upregulated protein synthesis and proteasome degradation in recurrent tumors. (**a**) Gene set enrichment analysis (GSEA) demonstrated that gene sets related to protein translation and rRNA processes were upregulated in recurrent PDX tumors compared with primary tumors. (**b**) Visualization of the significantly enriched pathway (FDR < 0.05) in recurrent PDX tumors compared with the pathways in primary tumors. (**c,d**) RNA-seq heatmap indicated increased expression of the Reactome translation gene set (**c**) and proteasome-encoding genes in recurrent patient samples (**d**). (**e**) mRNA expression levels of proteasome-encoding genes in human ATRTs (*n* = 24) were higher than those in normal brain tissues (*n* = 4). These data are presented in the box-and-whisker plot (min–max values), * *p* < 0.05, ns: nonsignificant, Mann–Whitney U test. (**f**,**g**) Comparison of IHC staining for PSMD4 (**f**) and PSMB4 (**g**) in normal brain tissues and the patient tumor samples. The patient’s recurrent ATRTs exhibited higher levels of PSMD4 (**f**) and PSMB4 (**g**) compared with the primary ATRTs and normal brain tissues. Scale bar, 50 µm.

**Figure 3 cancers-12-00752-f003:**
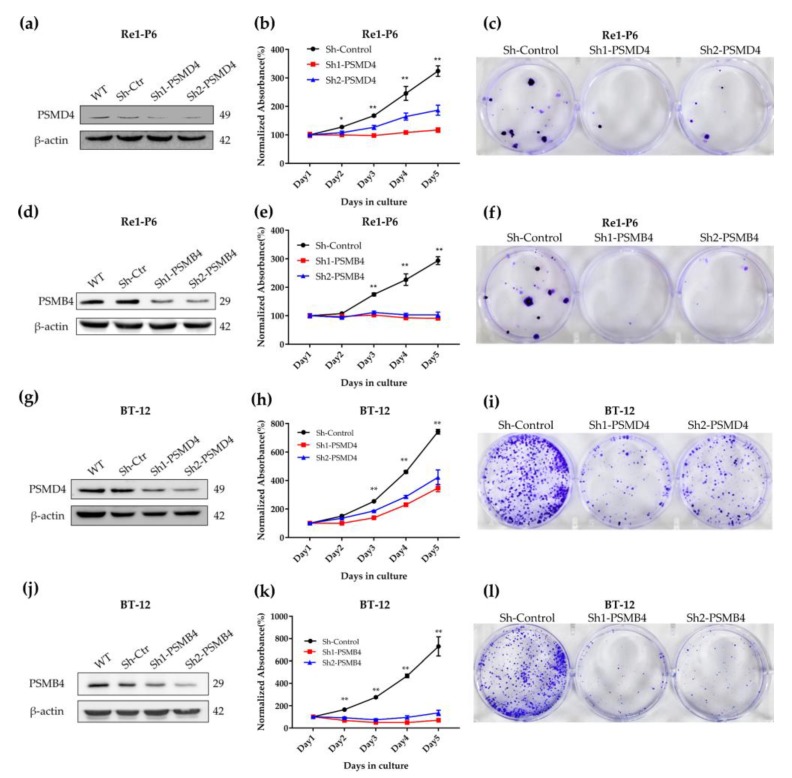
PSMD4 and PSMB4 are essential for ATRT cell survival. (**a,d,g,j**) Immunoblotting for PSMD4 (**a**,**g**) and PSMB4 (**d**,**j**) in Re1-P6 and BT-12 cells. After 48 h of transfection with scramble shRNA, shRNA-*PSMD4* or shRNA-*PSMB4*, the cellular extracts were subjected to Western blotting with anti-PSMD4 and anti-PSMB4. (**b,e,h,k**) Proliferation assays were performed on Re1-P6 and BT-12 cells after transfection with scrambled shRNA, shRNA-*PSMD4* or shRNA-*PSMB4*. (**b**,**e**,**h**,**k**) MTT assay revealed decreased expression of PSMD4 or PSMB4, resulting in decreased proliferation of Re1-P6 cells (**b**,**e**) and BT-12 cells (h,k). * *p* < 0.05, ** *p* < 0.01, Mann–Whitney U test. Data are presented as the mean ± standard deviation (*n* = 5 wells). (**c**,**f**,**i**,**l**) Low clonogenic capacity in PSMD4 down-expression (**c**,**i**) and PSMB4 down-expression (**f**,**l**) ATRT cells.

**Figure 4 cancers-12-00752-f004:**
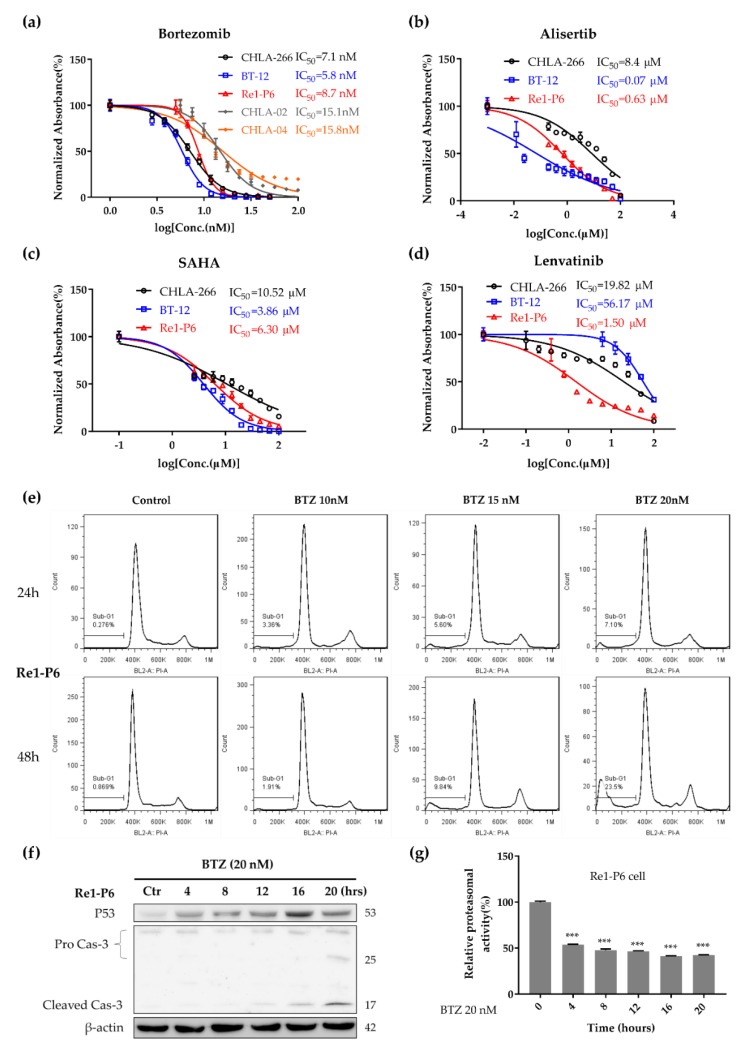
Proteasome inhibitor bortezomib (BTZ) inhibited proliferation and induced apoptosis in Myc-ATRT cells. (**a**) IC_50_ values of BTZ against 5 ATRT cell lines (CHLA-266, BT-12, Re1-P6, CHLA-02 and CHLA-04), (**b**–**d**) IC_50_ values of Aurora A kinase inhibitor alisertib (**b**), HDAC inhibitor SAHA (**c**) and lenvatinib (**d**) in three Myc-ATRT cell lines. MTT and MTS proliferation assay was performed 72 h after exposing cells to the drugs. (**e**) Cell cycle analysis in BTZ-treated Re1-P6 cells. Re1-P6 cells were incubated with BTZ (10, 15 and 20 nM) for 24 and 72 h. BTZ dose- and time-dependently increased the sub-G1 population. (**f**) Immunoblotting for p53 and cleaved caspase 3 in BTZ-treated Re1-P6 cells. Re1-P6 cells were incubated with BTZ (20 nM) for indicated durations. BTZ induced the accumulation of cleaved caspase 3 and p53 in BTZ-treated cells. (**g**) BTZ inhibited proteasome activity in Re1-P6 cells. Whole-cell lysates were incubated with substrate Suc-Leu-Leu-Val-Tyr-7-Amino-4-methylcoumarin at 37 °C for 2 h. The released fluorescence was quantified using a 380/460-nm filter. *** *p* < 0.001, *t*-test. The data are presented as the mean ± standard deviation (*n* = 3).

**Figure 5 cancers-12-00752-f005:**
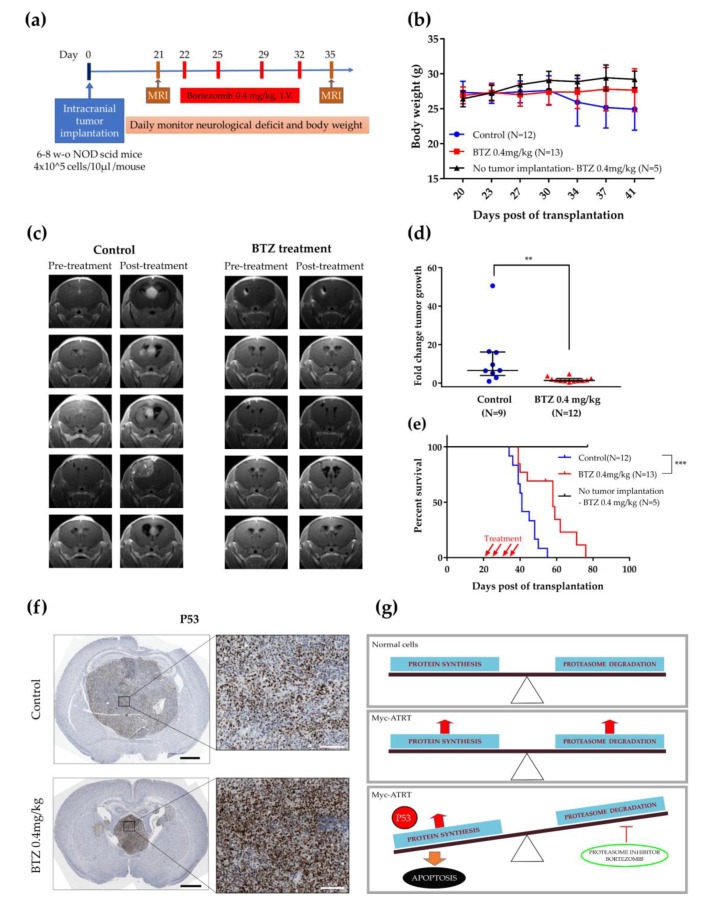
Antitumor effects of proteasome inhibitor BTZ in vivo. (**a**) In vivo experimental schema. Re1-P6 cells (4 × 10^5^ cells/10 µL) were injected into the right cerebral hemisphere of 6–8 week-old NOD.CB17-Prkdc^scid^/NcrCrl mice. The BTZ-treated group was administered four intravenous doses of 0.4 mg/kg BTZ at 22, 25, 29 and 32 dpt and the control group was treated with ddH_2_0 in 2% DMSO. (**b**) No significant weight loss was observed between the three groups (treated, control and toxicity-testing groups). Error bar, SD. (**c**) Representative pretreatment and posttreatment brain MRIs. BTZ significantly inhibited tumor growth in the treated group. (**d**) Quantification of tumor growth between the BTZ-treated group and the control group. ** *p* < 0.01, Mann–Whitney U test. The data are presented as the median ± interquartile range. (**e**) BTZ significantly prolonged the overall survival in the treated group (**** *p* < 0.0001, log-rank test). (**f**) Representative images of IHC staining for p53 show increased accumulation of p53 in the brain tumor of the BTZ-treated group. Scale bar, left panels: 1 mm and right panels: 100 µm. (**g**) Animation: concomitant upregulations of the proteasome load and capacity were observed in Myc-ATRT cells. Proteasome inhibitor BTZ inhibits tumor growth and induces apoptosis through the p53 signaling pathway.

**Figure 6 cancers-12-00752-f006:**
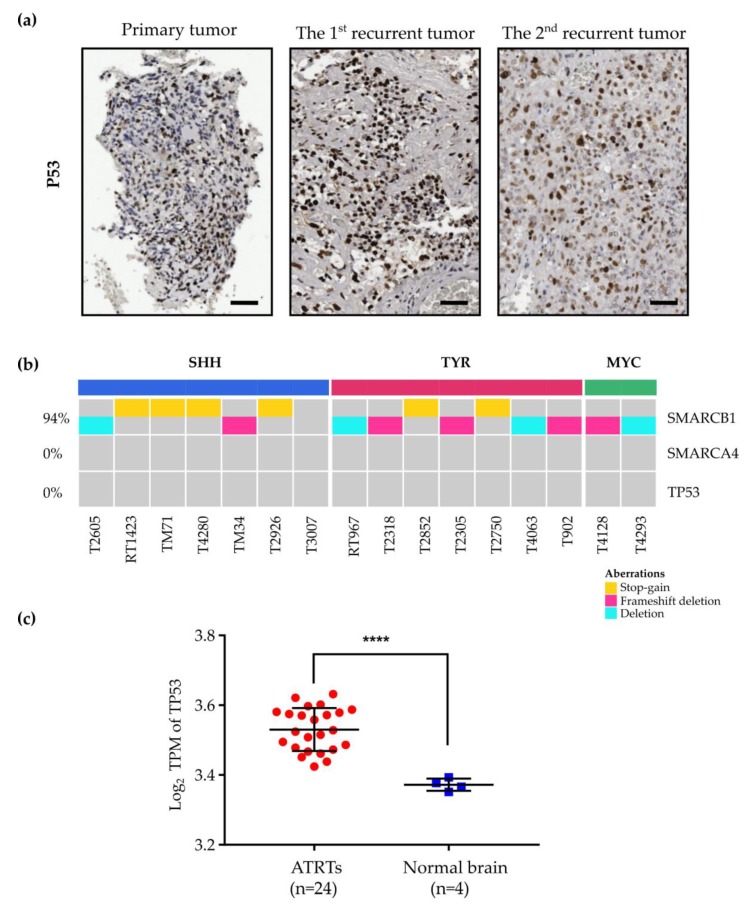
ATRTs are p53-proficient tumors. (**a**) IHC staining for p53 in primary and recurrent tumors. High expression of p53 in ATRTs. (**b**) WES of 16 patients with ATRT indicated no case of *TP53* mutation. (**c**) High expression of *TP53* mRNA in ATRTs (*n* = 24) compared with normal brain tissues (*n* = 4). **** *p* < 0.0001, Mann–Whitney U test. Scale bar, 50 µm.

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
