# Peer review of "Upregulation of Protein Synthesis and Proteasome Degradation Confers Sensitivity to Proteasome Inhibitor Bortezomib in Myc-Atypical Teratoid/Rhabdoid Tumors"

_cancers, 2020, doi:10.3390/cancers12030752_

Round 1

Reviewer 1 Report

Manuscript Cells-709270

Dear Editor,

I have reviewed the article “Upregulation of protein synthesis and proteasome degradation confers sensitivity to proteasome inhibitor bortezomib in Myc-atypical teratoid/rhabdoid tumors” by Huy Minh Tran T et al.  In my opinion, this paper deserves publication.

The main focus of this study was to investigate whether the upregulation of protein synthesis and proteasome degradation in Myc-ATRTs may be counteracted by the proteasome inhibitor bortezomib (BTZ). First, using primary and recurrent ATRT samples from an eight-month old infant, the Authors generated cell lines from primary, first recurrent and second recurrent PDX mice. Then, by performing gene expression analysis on matched primary and recurrent ATRT cells, They found that protein synthesis and the expression of proteasome components are increased in the recurrent tumors and that upregulation of the Myc pathway well correlates with protein synthesis and proteasome. Also, the Authors found that proteasome-encoding genes are highly expressed in ATRT tissues as compared with normal brain tissues and correlate with the malignancy of tumor cells, being essential for tumor cell survival. When the effects of the proteasome inhibitor bortezomib (BTZ) were investigated in vitro, BTZ was found to inhibit in a dose- and time-dependent manner the proliferation of three Myc-ATR cell lines and induce apoptosis. In vivo, BTZ inhibited tumor growth and prolonged survival of ATRT cells orthotopically implanted in nude mice. Based on these findings, the Authors conclude that BTZ may be a promising drug for targeted therapy of Myc-ATRTs.

Considering the lack of effective treatments for the Myc-ATRT patients, this topic is timely and may be potentially relevant to improve the clinical management of Myc-ATRTs. This paper deserves publication. However, there are some issues that have to be resolved in the opinion of this reviewer.

Minor Concerns

Results:

  1. This reviewer had some difficulty in understanding the results since some panels in the Figures are not described or, alternatively, are not cited in the text in a logic sequence: please revise Fig. 3, fig. 5, FigS3 and Fig.S6. Also, the Table 1 never was mentioned in the test.
  2. In the Figure 4, panel f, both full length and cleaved caspase 3 should be shown.

    Materials and Methods

    1. Methods include a variety of new technologies. To ensure reproducibility, this section should be revised in an effort to better describe the adopted experimental designs.
    2. Please include more information about the employed normal brain tissues (e.g. human or murine? Peritumoral or what?)
    3. Although BT-12 and CHLA-266 cells were provided by the Children’s Oncology Group, some information or references should be added.

      Discussion

      Regarding the possibility that targeting Myc may be a new strategy for treating children with atypical teratoid rhabdoid tumors, Alimova I.  and co-workers have recently published an interesting paper regarding the possibility to suppress tumor growth by inhibiting MYC transcriptomic program  (Alimova I. et al., Inhibition of MYC attenuates tumor cell self-renewal and promotes senescence in SMARCB1-deficient Group 2 atypical teratoid/rhabdoid tumors to suppress tumor growth in vivo. Int J Cancer. 2019 doi: 10.1002/ijc.31873. Epub 2019 Jan 10). Please discuss your findings in the light of this paper.

Reviewer 2 Report

The paper Tran et al established paired PDX xenocraft model from patient with ATRT. Characterization of the tumor indicates that in the recurrent tumor model, there is a Myc signature of gene expression, which suggest increased protein translation and degradation. They further describe and analyze the expression of proteasome and ribosome components in a broader sense by also extracting relevant data from several public databases. Next, the authors follow a sensible logic to postulate and test the myc-ATRT tumors should be sensitive for  proteasome inhibition.

Overall the paper is well written with the figures and data nicely presented. However, I have a major concern that the sensitivity to proteasome inhibitors or from proteasome subunit knock-down is only evaluated in the myc-ATRT cells, this does not allow for proper comparison of the ability to uniquely target these type of cancer cells.

Specific comments:

P.3 line 113 authors state that profile of 1 and 2 recurrent PDX highly similar to second recurrent patient sample. What is this based on? This does not seem apparent from the PCA in fig. 1d.

P.6 line 184. Figure 2 F and G claim more PSMD4 and B4 in primary and recurring tumors then normal CNS. No normal CNS is shown, how can this be determined from the data? Furthermore, mRNA data described earlier show no increase in proteasome subunits in the primary tumor. So that these protein data show similar levels in primary and recurrent tumors is not consistent with earlier mRNA expression analysis and confusing.

P.6 line 184 Why do authors conclude proteasome loading and capacity are more prominent? Levels of these proteins are higher, suggesting increased levels of proteasome… how much substrate is being loaded cannot be concluded from an IHC stain.

Figure 3. It is well known that proteasomes are essential, so knocking down two essential subunits will decrease proliferation in any cell if levels are sufficiently reduced. So, is this ATRT specific? Are myc-ATRT cells more sensitive and already affected by a more modest knock-down? As presented, I don’t see how this experiments reveals anything specific or unique for these cells. Since SSH-ATRT cells don’t upregulate transcription and proteasome, these should not be as sensitive to proteasome inhibitors and/or other control cells can be used.

Figure 4. The authors hypothesize that Myc-ATRT are more sensitive to proteasome inhibitor. However, in figure 4, I am missing the control cell lines to compare sensitivity with. Couldn’t the authors use SHH-ATRT as comparison and maybe also relate or discuss how myc-ATRT sensitivity compares with MM cells?

Figure S3: Interesting points is that the genes scoring around 0 are B8, 9, 10 (all three immunoproteasome actives site subunits that replace B6, B7, and B5 in the immunoCP). D5, D9,  are not a subunit but a chaperone involved in RP assembly. E1,2, 3, 4 and f1 can associate with CP when it replaces RP and are not involved in degradation of ubiquitinated proteins (or only as a hybrid form that has 1 RP).

Round 2

Reviewer 2 Report

In my previous comments I stated as main concern:"the sensitivity to proteasome inhibitors or from proteasome subunit knock-down is only evaluated in the myc-ATRT cells, this does not allow for proper comparison of the ability to uniquely target these type of cancer cells." 

Some other experiments in this context are apparently beyond the time frame provided by the journal according to the authors

"We are purchasing the normal human brain tissue for IHC staining of PSMD4 and PSMB4. The result is not available in this document due to the limitation of response time. Please give us three-week time for processing this data if it is absolutely required in this manuscript. Here, we would like to mention some data supporting our hypothesis that ATRTs, particularly recurrent tumors, highly expressed proteasome encoding genes, as compared with normal brain tissues."

or beyond the scope from this manuscript according to the authors.

"We agree that our data is not sufficient to answer the question of whether decrease in proliferation by knocking down PSMD4 and PSMB4 expression is specific for ATRTs. To answer this question, we will perform further study later in our subsequence research."

While  more comprehensive comparison of the tumor lines with other lines would be desired and I think would fall within the scope of the current manuscript as it provides important context and significance to the study, the authors did addressed a number of concerns and showed SHH-ATRT cells are less sensitive to proteasome inhibitors which at least in part supports their thesis. 

Round 3

Reviewer 2 Report

The authors adequately addressed the concerns and I have no further concerns on the manuscript.